# Use of 3′ Rapid Amplification of cDNA Ends (3′ RACE)-Based Targeted RNA Sequencing for Profiling of Druggable Genetic Alterations in Urothelial Carcinomas

**DOI:** 10.3390/ijms252212126

**Published:** 2024-11-12

**Authors:** Natalia V. Mitiushkina, Vladislav I. Tiurin, Aleksandra A. Anuskina, Natalia A. Bordovskaya, Ekaterina A. Nalivalkina, Darya M. Terina, Mariya V. Berkut, Anna D. Shestakova, Maria V. Syomina, Ekaterina Sh. Kuligina, Alexandr V. Togo, Evgeny N. Imyanitov

**Affiliations:** 1Department of Tumor Growth Biology, N.N. Petrov Institute of Oncology, 197758 St. Petersburg, Russia; nmmail@inbox.ru (N.V.M.); tyurinvladislav@gmail.com (V.I.T.); bordovskaya.n11@gmail.com (N.A.B.); ektrne@mail.ru (E.A.N.); terinadasha@gmail.com (D.M.T.); berkutv91@gmail.com (M.V.B.); anna.message19@gmail.com (A.D.S.); mvsyomina@gmail.com (M.V.S.); kate.kuligina@gmail.com (E.S.K.); a_togo@mail.ru (A.V.T.); 2Department of Medical Genetics, St. Petersburg Pediatric Medical University, 194100 St. Petersburg, Russia

**Keywords:** urothelial carcinoma, bladder cancer, *FGFR3*, *FGFR2*, *HER2*, fusion, mutation, amplification, NGS, targeted RNA sequencing

## Abstract

Targeted treatment of advanced or metastatic urothelial carcinomas (UCs) requires the identification of druggable mutations. This study describes the development of a 3′ Rapid Amplification of cDNA Ends (3′ RACE)-based targeted RNA sequencing panel which accounts for the status of all genes relevant to UC treatment, namely, *FGFR1-4*, *KRAS*, *NRAS*, *BRAF*, *ERBB2* (*HER2*), *CD274* (*PD-L1*) and *PIK3CA*. *FGFR2/3*-activating point mutations or fusions were found in 54/233 (23.2%) tumors. *FGFR3* rearrangements were identified in 11 patients, with eight of them being undetectable by commonly used PCR kits. In addition, one tumor contained a high-copy *FGFR2* gene amplification accompanied by strong overexpression of the gene. Mutations in *RAS/RAF* genes were present in 30/233 (12.9%) UCs and were mutually exclusive with alterations affecting *FGFR2/3* genes. On the contrary, activating events in the *HER2* oncogene (point mutations and overexpression), as well as *PIK3CA* mutations, which were relatively common, occurred with similar frequencies in *RAS/RAF*- or *FGFR2/3*-positive vs. negative samples. High *PD-L1* mRNA expression was associated with advanced disease stage and was not observed in tumors with increased *HER2* mRNA expression or in UCs with evidence for *FGFR2/3* activation. Three of the studied carcinomas had high-level microsatellite instability (MSI). Overall, more than half of the UCs had potentially druggable genetic alterations. The proposed NGS panel permits comprehensive and cost-efficient analysis of UC-specific molecular targets and may be considered in clinical routine.

## 1. Introduction

Urothelial carcinomas (UCs) comprise urothelial tumors of the upper tract and bladder malignancies. Upper urinary tract tumors, i.e., tumors originating in the renal pelvis or ureter, account for only 5–10% of all UCs [1]. Bladder cancer is the ninth most common cancer worldwide [2]. Its frequency is approximately four times greater in men than in women. UC is the most common histological subtype of bladder cancer, while other subtypes are rare entities [3]. Several UC-predisposing risk factors have been identified, including tobacco smoking, occupational exposures to aromatic amines, contamination of drinking water, etc. [1,2,4,5].

Genomic profiling of UCs provided evidence for inactivation of *TP53*, *ARID1A*, *KDM6A*, *KMT2D*, *CDKN2A/2B* and *RB1* tumor suppressor genes as well as activation of *FGFR3*, *CCND1*, *PI3KCA*, *ERBB2* and *MDM2* oncogenes [6,7,8,9]. The frequency of the involvement of the above genes varied significantly between studies, depending on the tumor location (upper tract UC or bladder carcinomas), the disease stage and various ethnic, geographic or lifestyle factors. Loriot et al. [9] found *TERT* promoter mutations in 77.5% of the studied samples, implying that *TERT* is probably the most commonly affected gene in metastatic UC.

Although, initially, most bladder carcinomas are diagnosed as non-muscle invasive bladder malignancies (NMIBCs), these tumors often recur and progress after initial treatment [4,5]. Muscle-invasive bladder carcinomas (MIBCs) are found in 20–25% of patients at the time of diagnosis, and around half of these patients later develop metastases. Metastatic bladder cancer is diagnosed in approximately 5% of all cases. Urothelial cancer of the upper urinary tract is more aggressive in comparison to bladder cancer: 60% of patients have invasive tumors at diagnosis [1]. The treatment of locally advanced or metastatic disease currently represents a major challenge. Until recently, platinum-based chemotherapy was the recommended first-line treatment option. A combination of enfortumab vedotin and pembrolizumab has recently become the treatment of choice for locally advanced or metastatic urothelial carcinoma [10,11,12]. According to the results of the phase III EV-302 trial, median progression-free survival (PFS) and overall survival (OS) were 12.5 months and 31.5 months, respectively, in patients receiving enfortumab vedotin plus pembrolizumab, compared to 6.3 months and 16.1 months, respectively, in patients receiving a standard platinum-based chemotherapy regimen [10]. Limited treatment options are currently available for the second and subsequent lines of therapy. One such option is therapy with erdafitinib, which is recommended for patients with susceptible *FGFR3* genetic alterations [11,12]. Other targeted agents can be used for the management of advanced UC; therefore, comprehensive characterization of potentially actionable molecular events is likely to increase the efficacy of UC treatment.

The 3′ Rapid Amplification of cDNA Ends (3′ RACE) is a long-known technology that is used for amplification and sequencing of the unknown 3′ parts of RNA molecules. The classic 3′ RACE protocol takes advantage of the presence of poly(A) sequences in mRNA molecules and allows for the study of their 3′ untranslated region (UTR) sequences [13]. The 3′ RACE method can be utilized for the identification of gene fusions involving a known partner gene sequence located at the 5′ end and the unknown partner gene sequence located at the 3′ end of the chimeric transcript [14]. This type of fusion is characteristic of the genes belonging to the fibroblast growth factor receptor (*FGFR*) family. *FGFR3* fusions are found with considerable frequency (2–6%) in UCs and can be targeted with the pan-FGFR inhibitor erdafitinib [6,15]. Genetic alterations in other *FGFR* family members are relatively uncommon; however, several studies have reported instances of *FGFR1* and *FGFR2* gene fusions in UCs [15,16].

We have recently described the 3′ RACE-based targeted RNA sequencing method, which allowed for simultaneous analysis of *FGFR* and other selected genes for activating mutations, gene fusions and changes in mRNA expression [17]. This method showed good performance in the study of biliary tract cancer due to low cost, simple and fast library preparation workflow, and the ability to identify a wide spectrum of clinically relevant alterations. In the current study, the same method was applied for the analysis of aberrations in *FGFR* family genes and other potentially druggable genetic events in a reasonably large consecutive series of UCs. Apart from demonstrating the usefulness of the above approach in molecular diagnostics of UC, this study attempted to define the frequency and analyze the co-occurrence of the clinically relevant molecular aberrations in urothelial cancer. In addition, all UCs were tested for *FGFR1-4* and *HER2* gene amplifications and microsatellite instability (MSI) using polymerase chain reaction (PCR) assays. 

## 2. Results

### 2.1. FGFR1-4 Gene Aberrations Identified by Targeted RNA Sequencing

The main clinical characteristics of 233 urothelial cancer patients are listed in Table 1. Formalin-fixed paraffin-embedded (FFPE) tissue samples from all patients were subjected to targeted RNA sequencing using the custom panel, as described in Section 4. The number of unique reads mapped to the regions of interest varied from 1550 to 162,081 per sample (median 38,049). The quality of results was assessed on the basis of the expression counts for the three referee genes (*DDX23*, *GOLGA5* and *SEL1L*), as described in Appendix B. Although there was no difference in the probability of discovering a point mutation between samples with the lowest and highest referee gene coverage, no gene fusions were identified in the 37 samples with the lowest referee gene counts, except for one case, where the *FGFR3::TACC3* gene fusion was supported by only a single read. Thus, it is possible that some gene fusions were missed in samples with poor RNA quality. 

Altogether, activating point mutations or fusions affecting *FGFR3* or *FGFR2* genes were found in 54/233 (23.2%) of all patients. Point mutations were identified in 44 tumors (*FGFR3:* p.S249C (*n* = 27), p.R248C (*n* = 5), p.Y373C (*n* = 8) and p.G370C (*n* = 5); *FGFR2:* p.Y375C (*n* = 1)). In three cases, *FGFR3* p.S249C mutation co-existed with an additional alteration in the same gene (either mutation p.R248C or p.G370C, or *FGFR3::TACC3* fusion). In addition, variants of unknown significance (*FGFR1* p.Q775E and *FGFR3* p.E664K) were identified in one patient each. These two variants were not considered in further analysis.

*FGFR3* rearrangements were found in 11 carcinomas (Table 2). *FGFR3::TACC3* fusions were identified in eight tumors, while other partner genes were involved in the remaining cases. All translocations involved exon 17 of the *FGFR3* gene. All tumors positive for *FGFR3::TACC3* fusion by next-generation sequencing (NGS) demonstrated the same alteration by PCR analysis. Notably, in all cases positive for the *FGFR3::TACC3* fusion, there was at least one alternative transcript with an intact *TACC3* reading frame (Table 2).

*FGFR2/3* mutations or fusions were evidently more frequent in upper tract UCs than in bladder carcinomas (10/21 (48%) vs. 30/147 (20%), *p* = 0.012), and in patients with localized disease compared to subjects with advanced or metastatic disease (29/99 (29%) vs. 16/9 9 (16%); *p* = 0.041). Patients with *FGFR2/3* mutations tended to be older (median: 69.5 years, range: 45–92), while subjects with *FGFR3* fusions tended to be younger (median: 63 years, range: 34–79) than patients without such alterations (median: 66 years, range: 20–87), although these differences were not statistically significant.

The level of mRNA expression of the *FGFR* family genes and other genes was assessed using NGS data, as described in Section 4 and in Appendix B. High *FGFR3* expression was characteristic for most tumors with *FGFR3* mutations (37/44, 84%). However, only 2/7 (29%) carcinomas with *FGFR3::TACC3* fusion (excluding a case with concomitant *FGFR3* mutation) had elevated *FGFR3* mRNA expression. At the same time, in all three cases, where *FGFR3* formed chimeric transcripts with genes other than *TACC3*, *FGFR3* mRNA expression was high. A single tumor with activating *FGFR2* p.Y375C mutation also demonstrated overexpression of the *FGFR2* mRNA.

Amplifications of the *FGFR* family genes were detected in several tumors using digital droplet PCR (ddPCR). Notably, high-level amplification (more than 100 copies) of the *FGFR2* gene, accompanied by *FGFR2* mRNA overexpression, was found in a 40-year-old woman with stage IIIA bladder UC. No co-occurring molecular events were detected in this patient. Seven UC samples had low-level amplification (from 3 to 5 copies) of the *FGFR2* gene; however, the increase in *FGFR2* mRNA expression was observed only in two of them, one of which was also *FGFR3* mutation-positive. *FGFR3* low-level amplifications were detected in two of the studied tumors, one of which had high *FGFR3* mRNA expression. There were two instances of the *FGFR1* gene amplification and one tumor with *FGFR4* gene extra copies; however, no increase in mRNA expression of the corresponding gene was observed in these samples.

### 2.2. Mutations in the Hot-Spot Regions of the RAS/RAF Genes

Overall, mutations in the *KRAS*, *HRAS*, *NRAS* and *BRAF* genes were found in 30/233 (12.9%) of the UCs studied with the 3′ RACE-based NGS method. Mutations in the *HRAS* gene were the most frequent, occurring in 13/233 (5.6%) patients. The spectrum of *HRAS* mutations included p.G12D (*n* = 3), p.G12S (*n* = 3), p.G13R (*n* = 2), p.Q61R (*n* = 2), p.K16T (*n* = 1), p.G12C (*n* = 1) and p.Q61L (*n* = 1) substitutions. Mutations in the *KRAS* gene were identified in 12/233 (5.2%) patients and were represented by p.G12D (*n* = 4), p.G12V (*n* = 3), p.G12C (*n* = 1), p.G13R (*n* = 1), p.Q61L (*n* = 1), p.K117N (*n* = 1) and p.A146T (*n* = 1) amino acid replacements. Mutations in the *NRAS* gene occurred in five patients: p.Q61K (*n* = 2), p.G13D (*n* = 1), p.Q61H (*n* = 1) and p.N67_R68 > IK (*n* = 1). One of the patients with *NRAS* mutation also had a concomitant mutation in the *HRAS* gene, and another patient had a concomitant mutation in the *KRAS* gene. *BRAF* mutations p.T599_V600insT and p.D594N were identified in one patient each.

*RAS/RAF* mutation-positive patients were younger than subjects without such mutations (median 62 years, range 20–81 vs. median 67 years, range 34–92; *p* = 0.007), but no association was found with gender, tumor location or stage of the disease.

### 2.3. HER2 (ERBB2) Aberrations in UCs: Point Mutations, Amplifications and mRNA Overexpression

Point mutations in the *HER2* gene were present in 17/233 (7.3%) samples. The following alterations were identified: p.S310F (*n* = 7), p.S310Y (*n* = 1), p.E348Q (*n* = 1), p.I370M (*n* = 1), p.I767M (*n* = 1), p.D769H (*n* = 1), p.G776I (*n* = 1), p.V777L (*n* = 1), p.L841V (*n* = 1), p.K854N (*n* = 1) and p.R868W (*n* = 1). Patients with *HER2* point mutations were older than subjects without such mutations (median 72 years (range 47–86) vs. median 66 years (range 20–92); *p* = 0.028). The presence of *HER2* point mutations in tumor samples was not associated with patients’ gender, tumor location or stage of the disease.

Digital droplet PCR detected an increase in the *HER2* copy number (CN ≥ 3) in 70/223 (31.4%) successfully analyzed samples. Tumors with high-level *HER2* amplification (CN ≥ 10) had markedly elevated *HER2* mRNA expression (Figure 1). There were no statistically significant differences in the *HER2* mRNA expression between groups with medium-level (5 ≤ CN < 10) or low-level (3 ≤ CN < 5) amplification and tumors without *HER2* amplification, although the former two groups contained a higher proportion of samples with *HER2* mRNA high expression (Table 3). The presence of a point mutation in the *HER2* gene was not associated with elevated *HER2* mRNA expression (*p* = 0.302). Patients whose tumors had increased expression of *HER2* mRNA were not statistically significantly different from other patients with respect to their age, gender, tumor location or stage of the disease.

### 2.4. Predictive Markers for Treatment with Immune Checkpoint Inhibitors: PD-L1 Expression and Microsatellite Instability (MSI)

*CD274* (*PD-L1*) mRNA expression was analyzed in all samples using targeted RNA sequencing data as described in Appendix B. The coverage of two fragments in the *PD-L1* gene was assessed: one of these fragments spanned exons 3 and 4 junction, and the other one spanned the junction between exons 5 and 6. The result was considered positive if both fragments demonstrated “high expression” or “overexpression” (see Appendix B for definition of the expression categories). If the results were discordant for the two fragments (e.g., “high” and “low” expressions), the analysis was considered not informative. By this analysis, 40 tumors were deemed *PD-L1*-positive, 181 UCs tested *PD-L1*-negative and 12 samples failed the analysis. 

PD-L1 immunohistochemistry (IHC) results were available for 55 of the analyzed UCs. Among them, 10 samples had PD-L1-positive staining in at least 1% of the tumor cells (Appendix A). Elevated *PD-L1* mRNA expression was found in 7/10 IHC-positive samples. One of these 10 samples failed *PD-L1* mRNA analysis, while the remaining two samples did not have increased *PD-L1* mRNA expression. Among 45 IHC-negative tumor samples, only two samples were *PD-L1*-positive at the mRNA level; the other two tumors failed the analysis, and the remaining ones were negative. Using PD-L1 status analysis by IHC as a reference method, we estimated the sensitivity of the 3′ RACE-based sequencing as 77.8%, and the specificity as 95.3%.

Patients with locally advanced or metastatic disease had *PD-L1*-positive tumors (determined by mRNA expression) more frequently compared to subjects with less advanced disease stages (23/92 (25%) vs. 11/94 (11.7%), *p* = 0.023), while no statistically significant associations were found for other analyzed parameters (age, gender, tumor location).

MSI-high status was revealed in only three out of 227 (1.3%) samples available for PCR and capillary electrophoresis.

### 2.5. PIK3CA Mutations in UCs

Different *PIK3CA* mutations were found in 42/233 (18%) UCs, including p.E545K (*n* = 18), p.E542K (*n* = 7), p.E726K (*n* = 2), p.H1047R (*n* = 2), p.R88L (*n* = 1), p.R88Q (*n* = 1), p.E110del (*n* = 1), p.N345T (*n* = 1), p.E365K (*n* = 1), p.E453_D454 > KK (*n* = 1), p.M1043I (*n* = 1), p.H1047L (*n* = 1), p.H1047Y (*n* = 1), p.R108L + p.R88Q (*n* = 1), p.E542K + p.R115L (*n* = 1), p.H1047R + p.E545K (*n* = 1) and p.H1047R + p.G106R (*n* = 1). Patients with mutations in the *PIK3CA* gene were slightly younger than subjects without such mutations (median 62.5 years, range 39–85 vs. median 67 years, range 20–92; *p* = 0.045), while there were no statistically significant associations with gender, tumor location or advanced stage of the disease.

### 2.6. Co-Occurrence of Potentially Actionable Molecular Alterations

Figure 2 represents the summary of all aberrations identified in 233 UCs. We analyzed the relationships between these genetic markers (Figure 3). *FGFR4* mRNA expression was excluded from the analysis because this gene generally had very low expression in UC samples and could not be measured with a sufficient level of confidence in a subset of tumors. 

*FGFR2/3* mutations or fusions appeared to be mutually exclusive with alterations of *RAS/RAF* genes. On the contrary, point mutations in the *HER2* gene occurred independently from the genetic events in the above genes. Also, elevated expression of *HER2* mRNA, with or without *HER2* gene amplification, did not show any relationship with *FGFR2/3* or *RAS/RAF* mutational status (i.e., occurred with similar frequency in mutation-positive and mutation-negative cases). Interestingly, high *PD-L1* mRNA expression was found to be mutually exclusive with high expression of the *HER2* mRNA, high expression of the *FGFR3* mRNA, or *FGFR2/3* point mutations or fusions (Figure 3).

## 3. Discussion

Locally advanced or metastatic urothelial cancer is a lethal disease with only a few currently available treatment options. In this study, known predictive and potentially targetable molecular alterations were investigated in 233 urothelial carcinoma samples using a recently developed 3′ RACE-based targeted RNA sequencing approach [17]. The list of studied genes included *FGFR1-4*, *KRAS*, *HRAS*, *NRAS*, *BRAF*, *HER2*, *PD-L1* and *PIK3CA*. 

Among the *FGFR* gene family members, aberrations affecting the *FGFR3* gene are the most common in UCs. The frequency of the *FGFR3* point mutations or rearrangements observed in this study (53/233, 22.7%) is very similar to estimates reported in other NGS studies [6,15]. Erdafininib is a targeted inhibitor of FGFR1-4 receptors, which is now approved for the treatment of locally advanced or metastatic UC with genetic alterations in the *FGFR3* gene [16,18]. The Therascreen FGFR RGQ RT-PCR kit (Qiagen, Hilden, Germany) was utilized as a companion diagnostic test in the erdafininib clinical trials. This test can identify four point mutations in the *FGFR3* gene (p.S249C, p.R248C, p.Y373C and p.G370C) and three *FGFR3* translocations (rearrangements between *FGFR3* exon 17 and *TACC3* exon 11 (v1) or exon 10 (v2), and *FGFR3::BAIAP2L1* fusion). This study showed a notably high diversity of *FGFR3::TACC3* rearrangements in UCs (Table 2), as was also shown previously for other tumor types [19,20]. Furthermore, *FGFR3* fusions involved three different partner genes. The Therascreen FGFR RGQ RT-PCR kit is potentially capable of identifying only 3/11 (27%) *FGFR3* fusions detected in this study. While the clinical phase II trial BLC2001 showed a relatively poor response rate in *FGFR2/3* fusion-positive tumors compared to cases with *FGFR3* point mutations (16% vs. 49%, respectively) [16], the subsequent phase III THOR trial demonstrated similar response rates in the above groups (44% vs. 47%, respectively) [18]. Although there is little information on the efficacy of erdafitinib in non-v1 *FGFR3* fusion-positive urothelial cancer, it is very likely that these rearrangements exert a comparable level of sensitivity to the targeted therapy. It is highly important to identify all UC patients who can benefit from erdafinib treatment; therefore, NGS testing (particularly RNA-based sequencing) is clearly superior to PCR as it is capable of revealing all *FGFR3* fusion variants.

Little is known about the factors which can modify the probability of response to erdafitinib treatment. In this study, it was shown that the activation of *HER2* via point mutation or increased expression is often coincident with *FGFR3* gene alterations (Figure 2). Also, the expression of the *FGFR3* gene was found to be up-regulated in most, but not all, tumors with *FGFR3* aberrations. These parameters deserve to be considered in future studies evaluating the efficacy of erdafitinib treatment.

Genetic events associated with FGFR2 receptor up-regulation are relatively rare in urothelial cancer. Several participants of the BLC2001 phase II erdafitinib trial had *FGFR2* gene rearrangements, which could be identified using the Therascreen FGFR RGQ RT-PCR kit (*FGFR2::BICC1* and *FGFR2::CASP7*) [16]. However, the phase 3 THOR trial did not include patients with *FGFR2* rearrangements, and the presence of *FGFR2* gene alterations in tumor tissue is no longer listed as an indication in the final FDA approval of erdafitinib. No *FGFR2* fusions were found in 233 urothelial tumors analyzed in this study. However, an activating p.Y375C mutation in the extracellular juxtamembrane domain of the FGFR2 receptor was identified in one sample. This mutation is known to represent a mutational hot-spot in cholangiocarcinoma, and, according to preclinical studies, is likely to be associated with sensitivity to FGFR inhibitors [21,22]. Another UC sample had an extraordinarily high level of *FGFR2* amplification (copy number, CN > 100), which was accompanied by significant overexpression of the gene. These observations confirm the involvement of the FGFR2 receptor activation in the pathogenesis of some UCs, thus warranting the use of FGFR inhibitors for this category of patients. On the contrary, *FGFR1* and *FGFR4* gene amplifications, being detected in several tumors, did not result in the concurrent increase in mRNA expression and, thus, are unlikely to represent true “driver” events.

*RAS/RAF* mutations were found in a significant proportion of studied patients (12.9%) and were mutually exclusive with the *FGFR2/3* alterations. Mutations in *HRAS* and *KRAS* genes occurred with similar frequency in UCs (5.6% and 5.2%, respectively). Currently, specific therapy is available only for certain cancers with the *KRAS* G12C mutation [23,24,25]. This substitution is rare in UC, with only a single positive tumor identified in the current study. At the same time, preclinical and early clinical studies have shown that *HRAS*-mutant cancers may be sensitive to the farnesyltransferase inhibitor tipifarnib [26,27,28,29]. *NRAS* and *BRAF* mutations are rare in urothelial cancer. In this study, two *BRAF* mutations (p.T599_V600insT and p.D594N) were identified in one patient each. p.D594N belongs to “class III” *BRAF* mutations, which result in decreased activity of this kinase and are not sensitive to therapy with BRAF inhibitors [30]. However, according to some reports, p.T599_V600insT mutation can be successfully targeted with dabrafenib and trametinib treatment [31,32].

*HER2* gene aberrations are common in UCs, according to previous reports [6,33]. These include point mutations and overexpression, which may or may not be related to the gene’s amplification. In this study, an increase in *HER2* gene copy number (≥3 copies) was detected in 31.4% of the studied samples using a digital droplet PCR test. However, the significant increase in *HER2* mRNA expression was characteristic only for tumors carrying high-level (≥10 copies) *HER2* gene amplification (Figure 1). *HER2* alterations (point mutations or increased expression) occurred with similar frequencies in samples with activating mutations in receptor tyrosine kinases *FGFR2/3*, in tumors with *RAS/RAF* gene mutations, and in *FGFR2/3-* and *RAS/RAF*-wild type carcinomas (Figure 2 and Figure 3). This is surprising given that, in other cancer types, *HER2* aberrations are mutually exclusive with activating events involving other receptor tyrosine kinases or *RAS/RAF* genes [34,35]. This observation should be taken into account when conducting sequential molecular diagnostics for UC, as the presence of *FGFR2/3* or *RAS/RAF* mutations does not exclude the concurrent activation of the HER2 receptor. Also, it should be further studied whether activation of the HER2 receptor can interfere with the efficacy of FGFR-targeted therapy.

HER2-directed therapies include anti-HER2 antibodies, small molecule inhibitors and antibody–drug conjugates. Clinical trials demonstrated generally poor performance of HER2-directed antibodies and small molecule inhibitors in urothelial cancer patients [33,36]. However, the MyPathway clinical trial revealed the promising efficacy of the pertuzumab + trastuzumab combination, although the number of participants with UC was small [37,38]. In that study, the treatment was shown to be effective predominantly in patients with *HER2* amplification and high level of expression (immunohistochemistry, IHC 2+/3+) but not in patients with *HER2* point mutations or concomitant *KRAS* mutations. HER2-directed antibody–drug conjugates belong to a relatively new class of drugs, which consist of a cytotoxic compound linked to the antibody directed against the cell surface molecules or receptors overexpressed by tumor cells. The advantage of such therapy is that it does not require the cancer cell to be strongly dependent on receptor signaling. Thus, it can be effective even in tumors that became resistant to the previous targeted treatment due to the activation of collateral pathways. Also, it does not require all tumor cells to express the receptor because the released cytotoxic compound can kill cells adjacent to HER2 expressors [39]. Recently, the HER2-directed antibody–drug conjugate trastuzumab deruxtecan was approved by the U.S. Food and Drug Administration (FDA) for the treatment of HER2-overexpressing (IHC 3+) cancers. The decision was based on the results of the DESTINY-PanTumor02 phase II trial, which included 41 bladder cancer cases [40]. Meanwhile, the same medication also received FDA approval for the treatment of patients with HER2-low (IHC 1+/2+) metastatic breast cancer based on the results of the DESTINY-Breast04 trial [41].

The important limitation of the current study is the absence of the HER2 assessment by IHC in the analyzed urothelial cancer samples. It remains the scope of future studies to examine the relationships between *HER2* mRNA and protein expression. Given the relatively frequent activation of the HER2 receptor in urothelial cancer and the emergence of new targeted treatment options for HER2-positive tumors, there is no doubt that routine diagnostic testing for HER2 expression status will become necessary for all UC patients in the near future. In addition, this study did not fully account for the proportion of tumor cells within the analyzed samples. Bulk RNA or DNA analysis, irrespective of whether it is performed by NGS or PCR, cannot provide reliable quantitative information for specimens with a low tumor cell fraction, e.g., it cannot consistently detect moderate changes in mRNA expression of a given gene.

This study evaluated *PD-L1* mRNA expression by the 3′ RACE-based NGS test. We were able to compare the *PD-L1* mRNA levels with the results of PD-L1 IHC for 55 urothelial carcinoma samples (Appendix A). Overall, the frequency of PD-L1 IHC-positive urothelial carcinomas (10/55, 18.2%) was found to be much lower than reported previously [42,43]. All five tumors with ≥ 10% PD-L1-positive tumor cells by IHC were found to have increased *PD-L1* mRNA expression (“positive” status) using the developed algorithm for NGS data analysis. However, only two out of five tumors with 1–5% of PD-L1-positive tumor cells were found to be “positive” by mRNA expression analysis. In addition, two IHC-negative samples in our study were found to express high levels of *PD-L1* mRNA, which can be attributed, for example, to the inability of diagnostic antibodies to detect some posttranslational modifications of the PD-L1 protein [44]. Intriguingly, high *PD-L1* mRNA expression in this study never co-occurred with high *HER2* mRNA expression in urothelial cancer samples (Figure 2). To our knowledge, such an observation has never been reported previously. Moreover, it is unlikely to be attributed to technical reasons because the same method demonstrated instances of concurrent *PD-L1* and *HER2* overexpression in biliary tract cancer carcinomas [17]. 

High PD-L1 expression in urothelial cancer was shown previously to be associated with advanced stage and poor prognosis, although some studies reported the opposite [45,46]. This investigation revealed an association between high *PD-L1* mRNA expression and advanced tumor stage. Although a correlation seems to exist between PD-L1 expression and the effectiveness of therapy with immune checkpoint inhibitors (ICIs), it is not strong: a significant proportion of PD-L1-negative tumors still benefit from such treatment [36,45]. Currently, the prescription of PD-L1 inhibitors in UC patients does not necessarily require PD-L1 expression analysis, and there is a need for more accurate predictive tests than PD-L1 IHC. At the same time, MSI-positive tumors are known to respond particularly well to ICI therapy [47]. However, MSI is rare in urothelial carcinomas [48]; in the current study, only 1.3% of tumors were found to be MSI-high. 

Different *PIK3CA* mutations were found in 42/233 (18%) UCs. Currently, *PIK3CA* alterations are not regarded as agnostic targets, and the use of PI3K signaling cascade inhibitors is limited to hormone receptor-positive metastatic breast malignancies [49].

## 4. Materials and Methods

### 4.1. Patients and Samples

This study included formalin-fixed paraffin-embedded (FFPE) tumor samples from 233 urothelial carcinoma (UC) patients, which were forwarded for molecular diagnostics to the N.N. Petrov Institute of Oncology (St. Petersburg, Russia) in the years 2022–2023. The inclusion criteria were (1) diagnosis of urothelial carcinoma, (2) availability of the primary tumor tissue and (3) patient’s informed consent. Each FFPE sample was studied by a pathologist to ensure at least 5% tumor cell content. All FFPE blocks were subjected to manual microdissection in order to maximize tumor cell content in pieces of tissues, which were subjected to nucleic acid extraction. Nucleic acids from FFPE tissue slices were extracted using a Quick-DNA/RNA MagBead kit (Zymo Research, Irvine, CA, USA). The results of tumor profiling were returned to the primary physicians of the corresponding patients; however, this study was not designed to examine how the UC characteristics influenced treatment decisions. This investigation was conducted in accordance with the Helsinki Declaration and was approved by the local ethics committee.

### 4.2. Targeted RNA Sequencing and Data Analysis

Next-generation sequencing (NGS) libraries were prepared with the 3′ rapid amplification of cDNA ends (3′ RACE)-based method as described previously [17]. Briefly, RNA was reverse-transcribed using random primers with the tail adapter sequence (5′-GTTCAGACGTGTGCTCTTCCGATCTNNNNNNNNNN-3′) (Figure 4). This tail is required for the anchored multiplex polymerase chain reaction (PCR) in order to perform the enrichment of libraries with sequences of interest. The custom panel included 77 primers targeting selected regions in the *FGFR1*, *FGFR2*, *FGFR3*, *FGFR4*, *KRAS*, *NRAS*, *HRAS*, *BRAF*, *ERBB2* (*HER2*), *CD274* (*PD-L1*) and *PIK3CA* genes (Appendix A). The libraries were PCR-amplified using indexed primers, subjected to concentration measurement and pulled together for NGS analysis. NGS was carried out either with the NextSeq 2000 instrument (Illumina, San Diego, CA, USA) or the GenoLab M device (GeneMind Biosciences, Shenzhen, China).

Bioinformatic analysis of the NGS data was performed according to the previously described procedure [17]. Briefly, the UMI-tools v1.1.4 package [https://github.com/CGATOxford/UMI-tools (accessed on 9 September 2024)] was used for the removal of PCR duplicates [50]. Although unique molecular identifiers (UMIs) were not systematically present in our adapter sequences, the distant part of the random primer used in the reverse transcription reaction provided enough mismatches due to the improper hybridization, and these mismatches were used as “preudo-UMIs” in the analysis [17]. The STAR v2.7.8a [51] software was utilized for read alignments. The improperly aligned reads were filtered out using the custom python script Filter_RACEbam.py [https://github.com/MitiushkinaNV/RACE_NGS (accessed on 9 September 2024)]. Another custom script, RACE_caller.py, was used for the variant calling. The called variants were annotated with the ANNOVAR software [https://annovar.openbioinformatics.org (accessed on 9 September 2024)] [52] and manually filtered, as described previously [17]. The reported exons and codons were numbered according to the Ensembl canonical transcripts [53]. The gene fusions were detected with the STAR-Fusion v1.10.1 package [54] and the pipeline used for the gene expression analysis is described in detail in Appendix B. R version 4.1.1 [55] was utilized for the normalization of gene expression counts and downstream analysis of expression data as described in Appendix B. The R package plot.matrix v1.6.2 [https://github.com/sigbertklinke/plot.matrix (accessed on 9 August 2024)] was used for data visualization.

### 4.3. Other Analyses

Digital droplet PCR (ddPCR) tests were developed for the analysis of *HER2* and *FGFR1-4* gene amplifications. A list of the primers and TaqMan probes is provided in Appendix A. ddPCR was carried out using a QX200 Droplet Digital PCR System (Bio-Rad, Hercules, CA, USA) according to the manufacturer’s instructions. The genes’ copy numbers were reported without correction for the tumor cell content, given that appropriate adjustments may require an additional high-precision pathological analysis of FFPE samples. 

The primers utilized for the PCR amplification and sequencing of different *FGFR3::TACC3* fusion variants are listed in Appendix A. The single TaqMan probe, containing the *FGFR3* exon 17 sequence, was used in real-time PCR for the detection of all tested *FGFR3::TACC3* variants. Real-time PCR reactions were carried out using the CFX96 instrument (Bio-Rad, Hercules, CA, USA). The PCR reaction mix consisted of 1-x PCR buffer (10 mM Tris-HCl, 50 mM KCl, pH 8.3), 250 mkM of each dNTP, 200 nM of each primer and probe and 1 U of TaqM polymerase (AlkorBio, St. Petersburg, Russia); the total concentration of Mg++ in the reaction was 2.5 mM and the reaction volume was 20 mkl. The PCR program included enzyme activation (95 °C, 10 min) and 45 amplification cycles (95 °C for 15 s, 65 °C for 30 s, 72 °C for 30 s). Sanger sequencing was performed using a CEQ 8000 Genetic Analysis System (Beckman Coulter, Brea, CA, USA) according to the manufacturer’s instructions.

Microsatellite instability (MSI) status was determined by PCR and capillary electrophoresis using a standard pentaplex panel, which included BAT25, BAT26, NR21, NR22 and NR24 mononucleotide markers, as described previously [56]. Ventana PD-L1 (SP263) Assay (Ventana Medical Systems, Oro Valley, AZ, USA) was utilized for the immunohistochemical assessment of PD-L1 expression in a subset of patients upon physician request as part of routine clinical diagnostics. A list of the gene names and underlying explanations is provided in Appendix A.

## 5. Conclusions

NGS is a preferred technology for molecular diagnostics of urothelial carcinomas. For example, unlike commercially available PCR kits, NGS is capable of detecting the entire spectrum of *FGFR3* fusions. Furthermore, multigene sequencing allows for the identification of some relatively rare genetic events, e.g., alterations in *FGFR2*, *KRAS*, *BRAF* and other genes. Importantly, RNA-based sequencing permits the analysis of mRNA expression of clinically relevant genes, like *HER2* (*ERBB2*) and *PD-L1*. However, additional studies that systematically compare the performance of NGS and clinically validated IHC assays are warranted. The 3′ RACE-based RNA sequencing procedure utilized in the current study is simple, fast and inexpensive. This method might be applied for a comprehensive assessment of druggable genetic alterations in patients with urothelial cancer. 

## Figures and Tables

**Figure 1 ijms-25-12126-f001:**
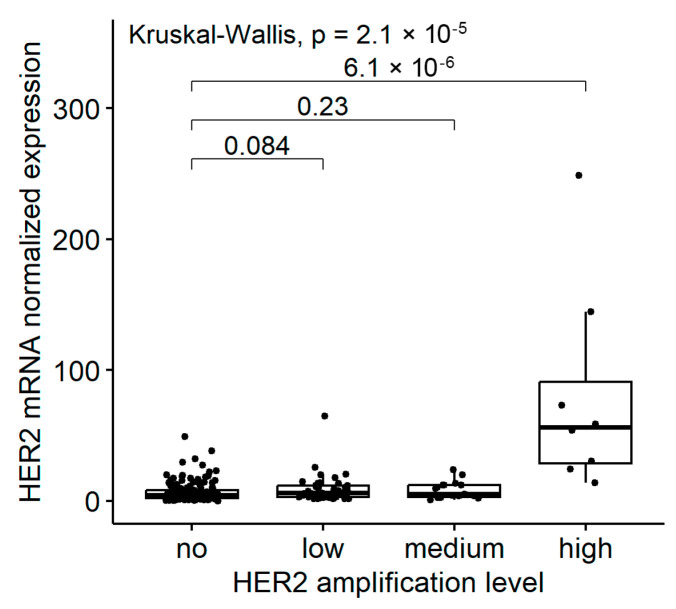
*HER2* mRNA expression, determined by targeted RNA sequencing, in urothelial carcinoma samples with different *HER2* gene amplification levels, determined by digital droplet PCR (ddPCR): no (copy number, CN < 3), low (3 ≤ CN < 5), medium (5 ≤ CN < 10) and high (CN ≥ 10).

**Figure 2 ijms-25-12126-f002:**
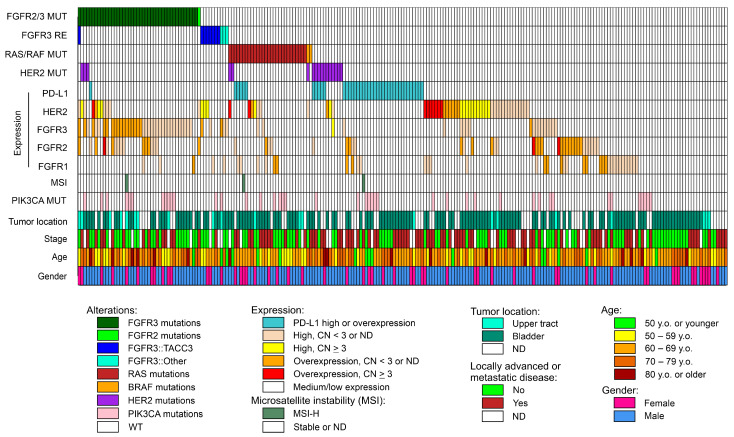
The graphical overview of all molecular aberrations identified in 233 urothelial carcinoma samples. Designations: MUT—point mutation; RE—rearrangement (fusion); WT—wild type; CN—copy number; ND—no data or not determined.

**Figure 3 ijms-25-12126-f003:**
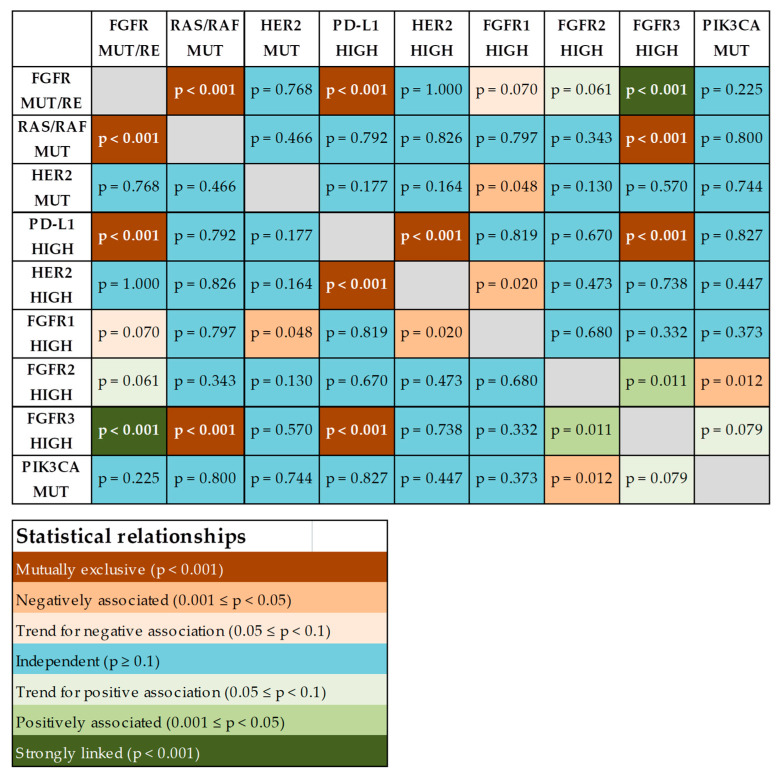
Associations between activating molecular alterations in urothelial carcinomas. Designations: MUT—point mutation; RE—rearrangement (fusion); HIGH—high expression/overexpression. The *p*-values were calculated using Fisher’s exact text.

**Figure 4 ijms-25-12126-f004:**
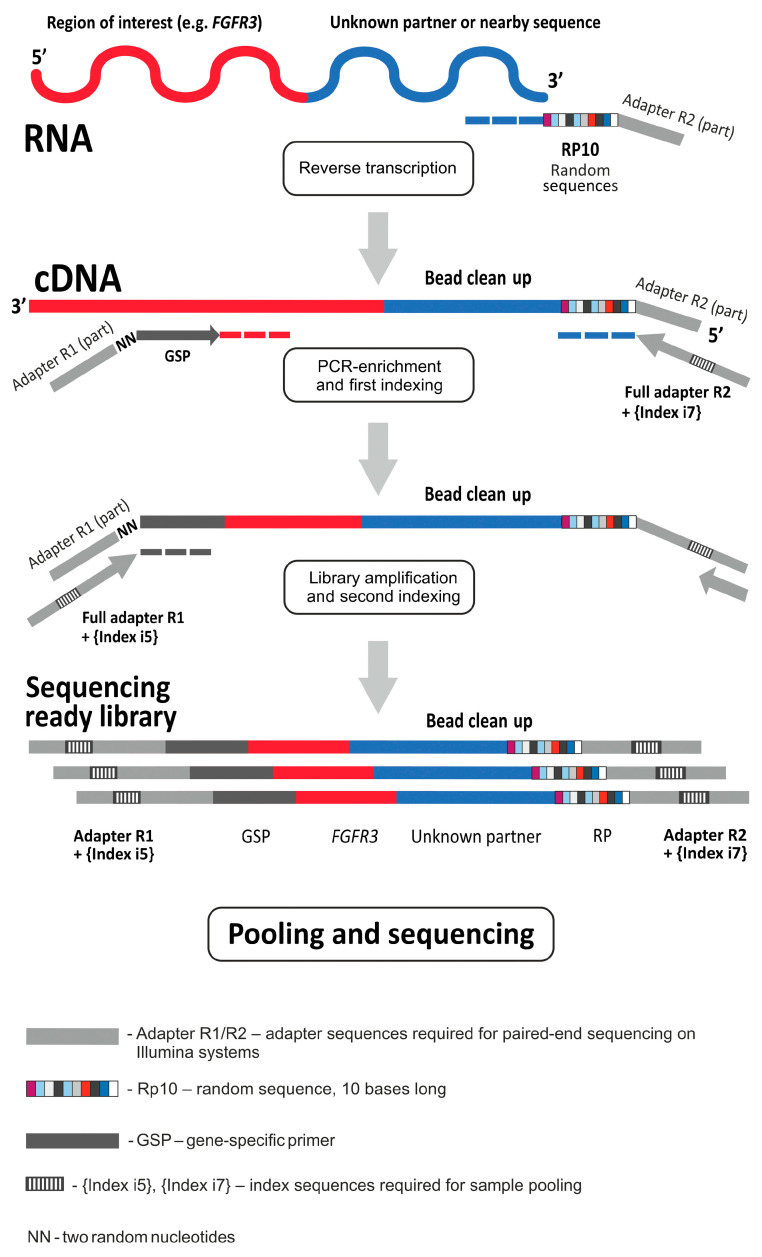
NGS libraries preparation using the 3′ RACE-based method.

**Table 1 ijms-25-12126-t001:** Main clinical characteristics of the studied urothelial cancer patients.

	Characteristic	Patients, *n* = 233
Age	Years, median (range)	66 (20–92)
Gender	Male, *n* (%)	176 (75.5%)
	Female, *n* (%)	57 (24.5%)
Tumor location	Upper tract, *n* (%)	21 (9.0%)
	Bladder, *n* (%)	147 (63.1%)
	ND, *n* (%)	65 (27.9%)
Locally advanced or metastatic disease (stage IIIB or IV)	Yes, *n* (%)	99 (42.5%)
	No, *n* (%)	99 (42.5%)
	ND, *n* (%)	35 (15.0%)

ND—no data.

**Table 2 ijms-25-12126-t002:** FGFR3 fusions identified in the current study.

Patient	#Alternative Transcript	5′ Partner Gene	5′ Ensembl Transcript ID	5′ Partner Exon	5′ Insertion of the Intronic Sequence	3′ Partner Gene	3′ Ensembl Transcript ID	3′ Partner Exon	3′ Insertion of the Intronic Sequence	3′ Gene Reading Frame	Number of Supporting NGS Reads	*FGFR3* mRNA Expression Level ^α^
#77	1	*FGFR3*	ENST00000440486	17		*TACC3*	ENST00000313288	4	−1591_−1512 ^β^	INFRAME	192	medium
#147	1	*FGFR3*	ENST00000440486	17		*TACC3*	ENST00000313288	5		FRAMESHIFT	6	high
	2	*FGFR3*	ENST00000440486	17	+1_+49	*TACC3*	ENST00000313288	5	−1535_−1481 ^β^	INFRAME	43	
#58	1	*FGFR3*	ENST00000440486	17		*TACC3*	ENST00000313288	8		INFRAME	186	medium
#211	1	*FGFR3*	ENST00000440486	17		*TACC3*	ENST00000313288	11		INFRAME	20,387	overexpression
#30	1	*FGFR3*	ENST00000440486	17		*TACC3*	ENST00000313288	11		INFRAME	1 ^γ^	medium
#52	1	*FGFR3*	ENST00000440486	17	+1_+3	*TACC3*	ENST00000313288	11	−742_? ^δ^	?	2247	overexpression
	2	*FGFR3*	ENST00000440486	17		*TACC3*	ENST00000313288	11		INFRAME	722	
#9	1	*FGFR3*	ENST00000440486	17	+1_+32	*TACC3*	ENST00000313288	12	−39_−1	INFRAME	129	medium
	2	*FGFR3*	ENST00000440486	17		*TACC3*	ENST00000313288	12		FRAMESHIFT	2	
#2	1	*FGFR3*	ENST00000440486	17		*TACC3*	ENST00000313288	14		INFRAME	48	medium
	2	*FGFR3*	ENST00000440486	17	+1_+25	*TACC3*	ENST00000313288	14	−2987_? ^δ^	?	5	
#106	1	*FGFR3*	ENST00000440486	17		*ADD1*	ENST00000683351	3		INFRAME	5306	high
#130	1	*FGFR3*	ENST00000440486	17		*SMIM14*	ENST00000295958	3		INFRAME	4411	overexpression
#6	1	*FGFR3*	ENST00000440486	17		*UACA*	ENST00000322954	14		INFRAME	70	high

^α^ The expression categories were defined as described in Appendix B. ^β^ The exact coordinates of the inserted sequences in these cases were determined using Sanger sequencing. ^γ^ The presence of the fusion in this sample was confirmed by PCR test. ^δ^ The insertion sequence could not be determined by Sanger sequencing because the fragment containing insertion failed to be amplified by PCR. Abbreviations: NGS—next-generation sequencing; PCR—polymerase chain reaction.

**Table 3 ijms-25-12126-t003:** The *HER2* mRNA expression in tumors with various *HER2* gene copy numbers.

*HER2* Gene Amplification	*HER2* mRNA Expression Level ^α^			Total
	Low or Medium	High	Overexpression	
High-level (CN ≥ 10)	0	1 (12.5%)	7 (87.5%)	8
Medium-level (5 ≤ CN < 10)	10 (58.8%)	6 (35.3%)	1 (5.9%)	17
Low-level (3 ≤ CN < 5)	31 (68.8%)	12 (26.7%)	2 (4.4%)	45
No amplification (CN ≤ 3)	123 (80.4%)	23 (15.0%)	7 (4.6%)	153
ND	7 (70.0%)	1 (10.0%)	2 (20.0%)	10

^α^ The expression categories were defined as described in Appendix B. Abbreviations: CN—copy number; ND—not determined.

## Data Availability

The data that support the findings of this study are available from the corresponding author upon reasonable request.

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
