# Peer review of "Use of 3′ Rapid Amplification of cDNA Ends (3′ RACE)-Based Targeted RNA Sequencing for Profiling of Druggable Genetic Alterations in Urothelial Carcinomas"

_ijms, 2024, doi:10.3390/ijms252212126_

Round 1

Reviewer 1 Report

Comments and Suggestions for Authors

·         Avoid using abbreviations in the title (RACE 3'; 3' rapid amplification of cDNA ends, NGS;  next-generation sequencing).

·         I think it would be good if authors could add more methodological/results details in the abstract (e.g., number/age of patients, date of sampling took place, place where the study was conducted, data analysis, significant differences).

·         Abbreviations should be defined at first mention in each section (e.g., 3' RACE, KRAS, FFPE…etc.). A list of abbreviations should be also placed at the end.

·         Please used “patients” instead of “cases” throughout. Was this a case-control study?

·         The introduction could be improved. Authors do not have the strong-enough biological background to review the molecular biology of UCs. There is also need to provide more details about 3' RACE. For example, What is 3' RACE? Can cDNA be amplified? How to amplify a gene from cDNA? Also, the research gap and the associated novelty/contribution aspect of the paper are very weak (Line 58-64). What is new? Why this study is important in light of previous studies.

·         Line 385-391: Please clearly describe the selection of participants, including eligibility and exclusion criteria.

·         Line 393: Referring to previous publications would be not enough to describe data analysis. For example, bioinformatic analysis should be described in much more details.

·         Line 380-382: The study comes to a weak conclusion. Authors draw conclusions without future implications. The conclusions would benefit from giving some consideration to how the paragraphs are structured and the thesis of each paragraph. Please include the conclusion in a separate section.

·         Please avoid using the terms “we” and “our” throughout (e.g., we developed, we applied, our data, our study).

Comments on the Quality of English Language

Minor language editing required.

Author Response

Comment: Avoid using abbreviations in the title (RACE 3'; 3' rapid amplification of cDNA ends, NGS; next-generation sequencing).

Response: We have now modified the Title to “Use of 3' Rapid Amplification of cDNA Ends (3' RACE)-based targeted RNA sequencing for profiling of druggable genetic alterations in urothelial carcinomas”.

Comment: I think it would be good if authors could add more methodological/results details in the abstract (e.g., number/age of patients, date of sampling took place, place where the study was conducted, data analysis, significant differences).

Response: We are sorry to say that this comment is difficult to meet: the Journal guidelines limit the size of the Abstract, therefore, the addition of more details will break this limit.

Comment: Abbreviations should be defined at first mention in each section (e.g., 3' RACE, KRAS, FFPE…etc.). A list of abbreviations should be also placed at the end.

Response: We provide the definitions of abbreviation upon first mention in the manuscript text. However, the Journal guidelines do not allow replicate these explanations in each section. The list of abbreviations is given at the end of the text. Also, we have added Supplementary Table S5 which explains gene names.

Comment: Please use “patients” instead of “cases” throughout. Was this a case-control study?

Response: Done.

Comment: The introduction could be improved. Authors do not have the strong-enough biological background to review the molecular biology of UCs. There is also need to provide more details about 3' RACE. For example, What is 3' RACE? Can cDNA be amplified? How to amplify a gene from cDNA? Also, the research gap and the associated novelty/contribution aspect of the paper are very weak (Line 58-64). What is new? Why this study is important in light of previous studies.

Response: We have added a few relevant paragraphs to the Introduction:

Genomic profiling of UCs provided evidence for inactivation TP53, ARID1A, KDM6A, KMT2D, CDKN2A/2B and RB1 tumor suppressor genes as well as activation of FGFR3, CCND1, PI3KCA, ERBB2 and MDM2 oncogenes [6,7,8,9]. The frequency of the involvement of above genes varied significantly between studies, depending on the tumor location (upper tract UC or bladder carcinomas), the disease stage and various ethnic, geographic or lifestyle factors. Loriot et al. [9] found TERT promoter mutations in 77.5% of the studied samples, implying that TERT is probably the most commonly affected gene in metastatic UC.

3' Rapid Amplification of cDNA Ends (3’ RACE) is a long-known technology, which is used for amplification and sequencing of the unknown 3’ parts of RNA molecules. The classic 3’ RACE protocol takes advantage of the presence of poly(A) sequences in mRNA molecules and allows to study their 3’ untranslated region (UTR) sequences [13]. The 3’ RACE method can be used for the identification of gene fusions involving known partner gene sequence located at the 5’ end and the unknown partner gene sequence located at the 3’ end of the chimeric transcript [14]. This type of fusions is characteristic for the genes belonging to fibroblast growth factor receptor (FGFR) family. FGFR3 fusions are found with considerable frequency (2-6%) in UCs and can be targeted with pan-FGFR inhibitor, erdafitinib [6,15]. Genetic alterations in other FGFR family members are relatively uncommon, however, several studies reported instances of FGFR1 and FGFR2 gene fusions in UCs [15,16].

We have recently described the (3' RACE)-based targeted RNA sequencing method, which allowed for simultaneous analysis of FGFR and other selected genes for activating mutations, gene fusions and changes in mRNA expression [17]. This method showed good performance in the study of biliary tract cancer due to low cost, simple and fast library preparation workflow, and the ability to identify a wide spectrum of the clinically relevant alterations. In the current study, the same method was applied for the analysis of aberrations in FGFR family genes and other potentially druggable genetic events in a reasonably large consecutive series of UCs. Apart from demonstrating the usefulness of the selected approach in molecular diagnostics of UC, this study attempted to define the frequency and analyze co-occurrence of the clinically relevant molecular aberrations in urothelial cancer. In addition, all UCs were tested for FGFR1-4 and HER2 gene amplifications and microsatellite instability (MSI) using polymerase chain reaction (PCR) assays. 

Comment: Line 385-391: Please clearly describe the selection of participants, including eligibility and exclusion criteria.

Response: We have now mentioned the inclusion criteria: “The inclusion criteria were: 1) diagnosis of urothelial carcinoma; 2) availability of the primary tumor tissue; 3) patient’s informed consent.” There was no exclusion criteria.

Comment: Line 393: Referring to previous publications would be not enough to describe data analysis. For example, bioinformatic analysis should be described in much more details.

Response: We have added the description of the bioinformatic analysis:  

Bioinformatic analysis of the NGS data was done according to the previously described procedure [17]. Briefly, the UMI-tools v1.1.4 package [https://github.com/CGATOxford/UMI-tools (accessed on 9 September 2024)] was used for the removal of PCR duplicates [50]. Although unique molecular identifiers (UMIs) were not systematically present in our adapter sequences, the distant part of the random primer used in the reverse transcription reaction provided enough mismatches due to the improper hybridization, and these mismatches were  used as “preudo-UMIs” in analysis [17]. The STAR v2.7.8a [51] software was used for read alignments. The improperly aligned reads were filtered out using the custom python script Filter_RACEbam.py [https://github.com/MitiushkinaNV/RACE_NGS (accessed on 9 September 2024)]. Another custom script, RACE_caller.py, was used for the variant calling. The called variants were annotated with the ANNOVAR software [https://annovar.openbioinformatics.org (accessed on 9 September 2024)] [52] and manually filtered, as described previously [17]. The reported exons and codons were numbered according to the Ensembl canonical transcripts [53]. The gene fusions were detected with the STAR-Fusion v1.10.1 package [54] and the pipeline used for the gene expression analysis is described in detail in Appendix 1. R version 4.1.1 [55] was utilized for the normalization of gene expression counts and downstream analysis of expression data as described in Appendix A. The R package plot.matrix v1.6.2 [https://github.com/sigbertklinke/plot.matrix (accessed on 9 August 2024)] was used for data visualization.

Comment: Line 380-382: The study comes to a weak conclusion. Authors draw conclusions without future implications. The conclusions would benefit from giving some consideration to how the paragraphs are structured and the thesis of each paragraph. Please include the conclusion in a separate section.

Response: We have extended the Conclusions section:

NGS is a preferred technology for molecular diagnostics of urothelial carcinomas. For example, unlike commercially available PCR kits, NGS is capable of detecting the entire spectrum of FGFR3 fusions. Furthermore, multigene sequencing allows identification of some relatively rare genetic events, e.g., alterations in FGFR2, KRAS, BRAF and other genes. Importantly, RNA-based sequencing permits the analysis of mRNA expression of the clinically relevant genes, like HER2 (ERBB2) and PD-L1. However, additional studies that systematically compare the performance of NGS and clinically validated IHC assays are warranted. The 3’ RACE-based RNA sequencing procedure utilized in the current study is simple, fast and inexpensive. This method can be applied for comprehensive assessment of druggable genetic alterations in patients with urothelial cancer.

Comment: Please avoid using the terms “we” and “our” throughout (e.g., we developed, we applied, our data, our study).

Response: We have re-phrased the text accordingly. Still, we retain a few phrases like mentioned above in those parts of the text, which are intended to emphasize the continuity of the described research. 

Reviewer 2 Report

Comments and Suggestions for Authors

This study presents a descriptive analysis of genetic alterations in selected genes potentially druggable in a cohort of 233 urothelial carcinomas. The authors found that 53 cases (23.2%) had alterations in FGFR3 or FGFR2, 30 cases (12.9%) had mutations in KRAS, HRAS, NRAS, or BRAF, and 17 (7.3%) had mutations in the HER gene. Additionally, 42 cases (18%) exhibited mutations in PIK3CA, and only 40 tumors (17%) were PD-L1 positive.

In my opinion, several key aspects are missing or require clarification:

1-Sample processing: There is insufficient detail about how the samples were processed. It was mentioned that at least 5% tumor cell content was ensured, but whether total RNA was extracted from whole tissue or only from the tumor area remains unclear. Discussing the quality of the sample and controls used would provide more clarity. 

2-More histological analyses are needed to identify altered cells with genomic alterations. It’s unclear why the cellular location was analyzed in only a small subset of cases, and why these particular cases were chosen.

3-Were the genomic findings used to personalize treatment? This is an important point that needs to be addressed.

4-Information about patient outcomes is missing and should be included.

5-Abbreviations should be explained when first mentioned. For instance, FFPE tissue, line 68, is not defined.

6-Table 1 needs clarification. For example, the Age section lists 66 samples, but there were 176 males and 57 females.  Please clarify.

7-Figure 2 is complex and could be made more accessible. Consider using Venn diagrams for a clearer presentation.

Author Response

Comment: Sample processing: There is insufficient detail about how the samples were processed. It was mentioned that at least 5% tumor cell content was ensured, but whether total RNA was extracted from whole tissue or only from the tumor area remains unclear. Discussing the quality of the sample and controls used would provide more clarity. 

Response: We have added a few relevant explanations:

…All FFPE blocks were subjected to manual microdissection in order to maximize tumor cell content in pieces of tissues, which were subjected to nucleic acid extraction. …

…this study did not fully account for the proportion of tumor cells within the analyzed samples. Bulk RNA or DNA analysis, irrespective of whether it is done by NGS or PCR, cannot provide reliable quantitative information for specimens with low tumor cell fraction, e.g., it cannot consistently detect moderate changes in mRNA expression of a given gene.

Comment: More histological analyses are needed to identify altered cells with genomic alterations. It’s unclear why the cellular location was analyzed in only a small subset of cases, and why these particular cases were chosen.

Response: We have used two complementary methods, i.e., NGS and PCR, in order to validate our findings with regard to analyzed genetic alterations. Histological methods are not sufficiently informative for studying mutations.

Comment: Were the genomic findings used to personalize treatment? This is an important point that needs to be addressed.

Response: We now address this issue in the “Materials and Methods”: “The results of tumor profiling were returned to primary physicians of the corresponding patients, however this study was not designed to examine how the UC characteristics influenced treatment decisions.”

Comment: Information about patient outcomes is missing and should be included.

Response: Unfortunately, we do not have information on patient outcomes.

Comment: Abbreviations should be explained when first mentioned. For instance, FFPE issue, line 68, is not defined.

Response: We have checked the text to ensure that all abbreviations are explained upon the first mention. The list of abbreviations is given at the end of the text. In addition, we have added Supplementary Table S5 which explains gene names.

Comment: Table 1 needs clarification. For example, the Age section lists 66 samples, but there were 176 males and 57 females.  Please clarify.

Response: 66 is a median age of the patients. We have modified this Table to improve its clarity.

Comment: Figure 2 is complex and could be made more accessible. Consider using Venn diagrams for a clearer presentation.

Response: The Venn diagrams can be used to represent co-occurrence rate of no more than two or three features. Thus, it is good for pairwise comparisons only. The pairwise associations are described in Figure 3. The Figure 2, though complex, allows representation and visual analysis of the entire studied molecular alterations. In this respect, it is similar to oncoplots which are frequently used to represent graphically the results of exome sequencing.

Round 2

Reviewer 1 Report

Comments and Suggestions for Authors

No further comments.

Author Response

Thank you!

Reviewer 2 Report

Comments and Suggestions for Authors

Although I miss the information about patient's outcomes and would like all this information to be used to personalize therapy, the paper is able to be published since it contains important information that could be useful for other authors and in the design of future clinical trials.

Author Response

Thank you!